# Validation of Inertial Sensors to Evaluate Gait Stability

**DOI:** 10.3390/s23031547

**Published:** 2023-01-31

**Authors:** Paul M. Riek, Aaron N. Best, Amy R. Wu

**Affiliations:** Ingenuity Labs Research Institute, Mechanical and Materials Engineering, Queen’s University, Kingston, ON K7L 3N6, Canada

**Keywords:** inertial measurement unit, gait stability, human locomotion, biomechanics

## Abstract

The portability of wearable inertial sensors makes them particularly suitable for measuring gait in real-world walking situations. However, it is unclear how well inertial sensors can measure and evaluate gait stability compared to traditional laboratory-based optical motion capture. This study investigated whether an inertial sensor-based motion-capture suit could accurately assess gait stability. Healthy adult participants were asked to walk normally, with eyes closed, with approximately twice their normal step width, and in tandem. Their motion was simultaneously measured by inertial measurement units (IMU) and optical motion capture (Optical). Gait stability was assessed by calculating the margin of stability (MoS), short-term Lyapunov exponents, and step variability, along with basic gait parameters, using each system. We found that IMUs were able to detect the same differences among conditions as Optical for all but one of the measures. Bland–Altman and intraclass correlation (ICC) analysis demonstrated that mediolateral parameters (step width and mediolateral MoS) were measured less accurately by IMUs compared to their anterior-posterior equivalents (step length and anterior-posterior MoS). Our results demonstrate that IMUs can be used to evaluate gait stability through detecting changes in stability-related measures, but that the magnitudes of these measures might not be accurate or reliable, especially in the mediolateral direction.

## 1. Introduction

Humans encounter a plethora of environmental conditions when walking outside, some of which must be countered to remain upright. Gait analysis has enabled scientific investigations of gait behavior, including the use of kinematic and stability measures. However, these studies have predominantly been conducted in controlled laboratory environments with the aid of optical motion-capture cameras. While the accuracy of these fixed systems has been well-established [1,2], it is challenging to apply these systems in environments that reflect the diversity of terrain one might experience in everyday life. This disconnect from real-world walking conditions represents a substantial limitation on walking stability and gait adaptation research.

One potential solution is to use inertial sensors to measure and assess gait. These wearable sensors do not directly measure position. Instead, they estimate movement from the fusion of sensor information from accelerometers, gyroscopes, and, sometimes, magnetometers within the inertial measurement unit (IMU). Several algorithms have been developed to estimate spatiotemporal parameters and kinematics from IMUs placed on one or more body segments [3,4,5]. These algorithms use some combination of filtering, drift correction, and biomechanical models to improve their accuracy [3,6], but their estimates may still differ from optical motion capture [5,7,8]. Several IMU configurations (and their associated algorithms) have been validated for use in measuring basic gait parameters, such as step length, step width, and gait-event detection [3,4,5,9]. Rebula et al. [3] found IMU stride length and duration values were within 1% of optical values but did not compare step width. Teufl et al. [5] compared the estimates of twelve different spatiotemporal measures between IMU and optical motion capture. They found that measures in the anterior–posterior (AP) direction could be measured much more accurately than those in the mediolateral (ML) direction. For example, the relative root-mean-square error (RMSE) of step length was 6.69% compared to 34.34% for step width. Although the large error might suggest that IMUs are not appropriate for estimating ML gait changes, it is unclear if IMU-based motion-capture systems have sufficient sensitivity to distinguish between walking conditions even with inaccurate estimates.

Few studies have determined whether IMUs are appropriate for evaluating more complex measures of gait stability [3,10,11,12]. These measures, such as the margin of stability (MoS), maximum Lyapunov exponent, and gait variability, require additional calculations and may, therefore, be more sensitive to position-estimation errors introduced by using IMU data. Guaitolini et al. [10] validated MoS determined by position estimates from IMUs calibrated with the aid of a camera system. They found that IMUs were able to accurately estimate MoS (median RMSE < 1 cm) and that accuracy was not affected by walking speed. Fino et al. [13] did not calculate MoS directly from IMUs, but instead showed that participant centripetal acceleration estimated from IMUs was strongly correlated ($R^{2}=0.72$) with MoS calculated from an optical motion-capture system. Several studies have also identified significant differences between conditions and groups using Lyapunov exponents calculated from IMU data [11,14]. Bruijn et al. [11] showed that Lyapunov exponents calculated using IMUs were highly correlated with exponents from optical data ($R^{2}>0.85$) and could be used to distinguish between walking speeds. Step variability has also been assessed using IMUs. Rebula et al. [3] reported estimates of step length and width variability were within 4% of optical values and detected differences between normal and eyes-closed walking. In contrast, Rantalainen et al. [12] found poor reliability for step-length variability (ICC = −0.23) and other variability parameters despite accurate mean values. These studies have primarily focused on assessing a single measure of stability or differences from a single condition. A more comprehensive assessment is needed to more broadly evaluate the suitability of IMUs for gait-stability analysis.

The aim of this study was to determine whether IMUs can be used to accurately evaluate gait stability. To perturb study participants and elicit different stability behaviours, we tested four walking conditions of Normal, Eyes Closed, Wide (approximately twice their normal step width), and Tandem (zero step width). We evaluated IMU-derived measures in three different ways to determine whether IMUs could differentiate among gait behaviours and to quantify their limits of agreement and reliability against measures derived from optical motion capture. Given IMUs’ past success in estimating gait kinematics and distinguishing among different walking conditions, we hypothesized that IMUs will be able to accurately differentiate among the four conditions. However, we expect that the IMU-derived measures will differ in magnitude from those calculated from the optical system, with higher differences for the ML measures than for the AP measures.

## 2. Materials and Methods

Healthy adult participants walked overground under four different conditions: Normal, Eyes Closed, Wide, and Tandem. Whole-body motion tracking was conducted simultaneously with an optical camera-based system and an IMU motion-capture suit to enable comparison of stability measures from individual walking strides captured by each system. We compared stability measures of margin of stability and maximum Lyapunov exponent and spatiotemporal measures of step width, step length, and their variability.

### 2.1. Experiment

Twelve healthy adults (N=12, 6 male and 6 female, age 23.8 ± 3.2 years, height 172.7 ± 10.3 cm, weight 71.7 ± 9.5 kg) participated in the study. Data from two participants were excluded from analysis due to equipment failure. All participants provided informed consent, and ethical approval was granted by the Queen’s University and Affiliated Teaching Hospitals Research Ethics Board.

Participants were instructed to walk normally (Normal), with their eyes closed (Eyes Closed), with approximately twice their normal step width (Wide), and with zero step width (Tandem). Two trials were conducted for each condition, and the eight total trials were randomly ordered. For each trial, participants walked twenty times back and forth along a path that was approximately 5.5 m in length at their self-selected speed. Strides related to turns at the end of each path were omitted, leaving an average of 85 steps per participant for each condition for analysis. The start of motion capture was synchronized for both the fixed camera and IMU systems to ensure that the same strides measured by each motion-capture system were selected for analysis. Prior to data collection, participants were given the opportunity to practice walking with the different conditions.

The IMU motion-capture suit (MVN Link, Xsens, Enschede, The Netherlands) was composed of seventeen IMUs placed on body segments (Figure 1). Lower body IMU locations were the sacrum and the thigh, shank, and foot of each leg. IMUs were also placed on the upper body at the scapula, upper arm, lower arm, and hand on each side, as well as the head and sternum. Software associated with the suit (Xsens MVN) was used to capture inertial data (obtained at 240 Hz and then downsampled to 120 Hz for analysis) and estimate gait kinematics of the participants [15]. The estimated whole-body center of mass (CoM) was based on the weighted average of each body segment. Xsens also estimated anatomical landmarks at the feet including `heel’ and `ball-of-foot’, which were used for gait-event detection and foot position. The IMU CoM and foot positions were filtered with a fourth-order Butterworth filter with a cut-off frequency of 6 Hz.

To compare IMU system measurements against an optical camera-based system (Vicon, Oxford, United Kingdom), participants were also outfitted with reflective markers. A total of eight markers were used in this study (Figure 1) and tracked at 100 Hz. The body CoM was estimated by averaging the position of markers on the left and right anterior superior iliac spine (ASIS) and posterior superior iliac spine (PSIS), and markers at the toe (2nd metatarsal head) and heel (calcaneus) of each foot were used to estimate foot position and gait events. Prior to analysis, marker positions were also low-pass filtered using a fourth-order Butterworth filter with a cut-off frequency of 6 Hz.

Heel-strike and toe-off gait events were found independently for each system. We assigned heel-strike and toe-off based on changes in the distance between the CoM and foot position in the forward walking direction [16]. This method was used for both systems and was comparable with the built-in Xsens foot contact detection.

### 2.2. Analysis

We calculated gait stability and spatiotemporal measures for comparisons between the two motion-capture systems. The measures we used included the minimum MoS and maximum Lyapunov exponent in the ML and AP directions. We also calculated walking speed, step length, step width, and step root-mean-square (RMS) variability. Walking direction was assigned as the vector from CoM position at the start of each gait cycle to CoM position at the end of each gait cycle.

MoS was defined as the distance between the extrapolated center of mass (XCoM) and the edge of the base of support (BoS) created by foot contact with the ground [17,18]. The XCoM accounts for both CoM position and velocity and was defined as
(1)$$\mathrm{XCoM}=\mathrm{CoM}+\frac{\dot{\mathrm{CoM}}}{\omega }$$
where CoM and $\dot{\mathrm{CoM}}$ are the position and velocity of the CoM, respectively, and ω is the eigenfrequency of the inverted pendulum model, defined as
(2)$$\omega =\sqrt{\frac{g}{L}}$$
where *g* is the acceleration due to gravity and *L* is the average leg length (standing height of the greater trochanter relative to the ground) of the participants.

The minimum MoS value was calculated for each gait cycle. To determine the minimum ML MoS, the ML edge of the BoS was estimated as the lateral foot position during single-leg stance. The marker on the second metatarsal head was used as the foot position for optical motion capture, while the estimation of the ball of the foot was used as the foot position for the IMU suit. The minimum AP MoS was estimated as the difference between the XCoM and the position of the leading foot at heel strike in the AP direction.

Maximum Lyapunov exponents provide a measure of how chaotic a system is and its sensitivity to initial conditions and have been shown to be correlated with fall risk [18]. Any time-series of kinematic data can be used to calculate a maximum Lyapunov exponent. For this experiment, we used the CoM velocity in AP and ML directions. Because Lyapunov exponents are dependent on the number of gait cycles included for analysis [19], the same number of cycles per trial was needed. We used the first 55 gait cycles as it was the minimum number of strides for all trials. We applied a version of the algorithm proposed by Rosenstein et al. [18,20,21]. The maximum divergence exponent was found by calculating the slope of the mean divergence curve from 0–0.5 strides, usually referred to as a short-term Lyapunov exponent.

We also calculated spatiotemporal parameters for comparisons between the two motion-capture systems, including walking speed, step length, step width, and their RMS variability. Walking speed was calculated from the average CoM velocity in the walking direction. Step length was the distance in the AP direction between the two feet at heel strike, and step width was the average ML distance between the two feet during stance. Optical step length and width were determined using the 2nd metatarsal marker, while the IMU step length and width were from the estimation of ’ball-of-foot’ position, matching the markers used for BoS calculation.

To account for differences in participant body size, kinematic and spatiotemporal measures were normalized using base units of standing leg length *L* and gravitational acceleration *g*. Step length, width, and MoS were normalized by leg length *L* (mean 0.92 m), and the CoM velocity was normalized by $\sqrt{gL}$ (mean 3.0 m/s) prior to Lyapunov exponent calculation.

We compared the effect of each condition (Normal, Eyes Closed, Wide, and Tandem) using three different methods: ANOVA testing, Bland–Altman, and intraclass correlation (ICC). The ANOVA enabled us to determine whether statistically significant differences among conditions would be the same between the two measurement systems. We performed the analysis on the ML and AP MoS, ML and AP Lyapunov exponent, step width, step length, and their variability. Repeated measures ANOVA was used with a significance level of α=0.05, followed by post-hoc t-tests with Holm–Sidak correction for multiple comparisons with Normal as the control condition [22].

Bland–Altman analysis was used to assess the agreement of measures determined by fixed motion capture with those determined by the IMU suit. This analysis was conducted for ML and AP MoS, as well as step length and width, on a stride-by-stride basis. Differences between measurements were calculated by subtracting IMU measurements from fixed motion-capture measurements of the same stride. The overall mean difference provided an estimate of the bias between measurement systems. The limits of agreement (LoA) were calculated as
(3)LoA=meandifference±1.96·SD(differences)
where SD(differences) is the standard deviation of the differences between measurements [23]. The LoA indicates the range where approximately 95% of future errors due to measurement system should fall, providing a measure of how consistently the two measurement systems agree regardless of bias [23].

We used the intraclass correlation coefficient (ICC) to assess reliability between the two methods. While Bland–Altman plots can assess the amount of measurement error between two different methods, the ICC assesses the ability to discern differences between steps from each condition despite measurement error. An ICC of less than 0.50 was considered poor reliability, between 0.50 and 0.75 as moderate, between 0.75 and 0.90 as good, and greater than 0.90 as excellent [24]. We calculated the ICC for ML MoS, AP MoS, step length, and step width using a two-way mixed-effects model with absolute agreement and a single measurement (A-1 method, [24,25]).

## 3. Results

In comparison to optical motion capture, we found that IMUs yielded differing estimates of gait measures, but these estimates still resulted in all but one of the same conclusions among walking conditions (summarized in Table 1). The differences between IMU and Optical were primarily found in the ML direction, with relatively poorer estimates of step width and ML MoS than for AP measures. ICC analysis also revealed that assessing each condition separately further exacerbated the differences between IMU and Optical in comparison to evaluating the strides from all conditions together (Table 2).

Study participants walked under the four conditions of Normal, Eyes Closed, Wide, and Tandem at their self-selected speeds. Compared to Normal, walking speed only differed for the Tandem condition, with participants choosing to walk about 12% slower on average (IMU p=0.012, Optical p=0.011). Qualitatively, the mean left and right foot position over a stride had noticeable differences between IMU and Optical, but similar trends among the four conditions (Figure 2). Normal and Eyes Closed exhibited similar step width while the Wide condition had larger step widths and the Tandem had zero or negative step width. In contrast, XCoM and CoM trajectories seemed more aligned between IMU and Optical for all four conditions. XCoM and CoM range increased for the Wide condition and decreased in the Tandem condition. The small differences in XCoM trajectories between Optical and IMU were also generally consistent for all conditions, but slightly larger for the Tandem condition. IMU and Optical data also appeared to be comparable over the gait cycle in the AP direction (Appendix A Figure A1).

Both IMU and Optical data showed the same significant differences among conditions for all analyzed stability measures, except for step width variability (Figure 3). Compared to Normal, differences were found in the ML direction for MoS, the short-term maximum Lyapunov exponent, and step width for Tandem and Wide (post-hoc paired t-tests p<0.05, Figure 3A). For the AP direction, the MoS was significantly different from Normal for the Tandem conditions (IMU p=0.0013, Optical p=0.0024), as well as for the Lyapunov exponent and step length variability for Eyes Closed and Wide (p<0.05, Figure 3B). Different conclusions between Optical and IMU data were only indicated for step width variability of the Wide condition, which was significantly different from Normal for Optical (p=0.010) but not for IMU (p=0.054).

The Bland–Altman plots revealed some bias for each of the four measures. The positive overall mean difference of 0.008 and 0.035 illustrated that the IMU captured smaller ML MoS and narrower step widths, respectively, than the Optical system (Figure 4A,C). In contrast, a negative overall mean difference of −0.054 and −0.008 showed that IMUs measured greater AP MoS and longer step lengths than Optical (Figure 4B,D). Qualitatively, mean differences were generally centered around the overall difference, suggesting that the biases were not correlated to the magnitude of the measurement values. The 95% limits of agreement were 2.9 times wider for step width (0.25) compared to step length (0.086), suggesting worse agreement. The limits of agreement were also slightly wider for ML MoS (0.14) compared to AP MoS (0.13), although AP MoS had a larger bias in magnitude.

ICC analysis showed moderate to excellent reliability when all the conditions were combined, but relatively poorer reliability for some of the individual conditions (Table 2, Figure 5). Combining all four conditions increased the variance of the data and resulted in higher ICC values. ML and AP MoS showed moderate reliability (ICC 0.74 each), and step width and length showed excellent reliability (ICC 0.91 and 0.93, respectively). However, individual conditions performed poorly. ML MoS showed poor reliability for each individual condition (mean ICC 0.31), while AP MoS showed moderate to good reliability (mean ICC 0.71). Step width showed poor reliability, except for the Wide condition (ICC 0.83 for Wide, mean ICC 0.48 for others), and step length showed good to excellent reliability for each condition (mean ICC 0.92). Averaging over all conditions, AP MoS and step length had the highest ICC values, while ML MoS and step width performed the worst.

## 4. Discussion

We aimed to validate the use of IMUs for measuring gait stability and hypothesized that IMU stability measures would differ in magnitude from Optical ones but could still be used to differentiate among different walking conditions. As expected, IMU-derived values differed from Optical, with positive differences found for ML MoS and step width and negative differences for AP MoS and step length. Step width also exhibited the largest limits of agreement followed by ML MoS. Similarly, the poorest reliability was found for ML MoS and step width when considering the four different conditions separately. Despite these differences, the use of IMU data resulted in nearly identical between-conditions conclusions as Optical, suggesting that IMUs are capable of accurately capturing trends in walking stability for MoS, Lyapunov exponents, step width, step length, and step length variability.

The accuracy and reliability of measuring ML parameters with IMUs was weaker compared to their AP counterparts (Figure 4 and Figure 5), which agrees with previous research [3,5,10]. One possible reason is that the calculation of step width requires accurate reconstruction of the entire lower-body kinematic chain, and therefore the measure might be more sensitive to the quality of the calibration procedure and segment length scaling of the underlying biomechanical model [5]. In contrast, optical motion-capture systems can directly measure the distance between two markers and, thus, provide a direct measurement of step width. Optical systems can also track markers to sub-millimeter accuracy [2], although larger errors of the order of centimeters can be introduced by factors such as marker placement, CoM estimation methods, and soft-tissue artifacts [26]. Similarly, sources of error with inertial motion capture include accelerometer and gyroscope measurement errors recorded by individual IMUs, along with possible inaccuracies introduced by filtering, body-segment scaling, and calibration to estimate position [27].

In comparing Eyes Closed, Wide, and Tandem against the Normal condition, measures derived from IMUs delivered the same conclusions as from Optical except for one measure. The same significant pairs were found for ML and AP MoS, ML and AP Lyapunov exponent, step width, step length, and step length variability, but not for step width variability (Figure 3). Post-hoc analysis found no significant difference in pair-wise comparisons for the IMU, but group-wise comparisons did reveal that step width variability varied with conditions for both IMU (p=0.039) and Optical (p=0.030). Thus, we were able to detect some difference in step width variability among conditions, but, with IMUs, could not conclude for which conditions in particular.

Previous studies have used optical motion capture to detect differences between populations of the order of a few centimeters for step width (∼7 cm, [28]), ML MoS (∼3 cm, [29]), and AP MoS (∼5 cm, [30]), which is much smaller than the range of Bland–Altman LoA of 23 cm, 13 cm, and 12 cm, respectively. As such, absolute agreement of the two measurement systems could be characterized as poor for these applications. The step width Bland–Altman plot (Figure 4C) had several clusters of data, which were found to correspond to specific participants. This clustering suggests that there may be a somewhat consistent measurement system bias for each participant. For example, Bland–Altman plots for individual participants would yield one participant with an overall mean difference of +0.16 for step width and another with −0.06, with both having limits of agreement within 0.03 of their overall mean. This participant-specific bias could be due to calibration or sensor placement errors, or due to individual differences in walking style. The procedure we used for sensor placement and calibration was the same for all participants, but accuracy of the IMU estimates could be sensitive to subtle changes that may be difficult to detect or control, especially when using proprietary software.

There were several limitations to our study. Only healthy adult participants were included in this study, and it is possible different conclusions would be drawn with other populations, such as individuals with pathological gait. The experiment was also conducted on a flat indoor surface, possibly facilitating better sensor drift correction in comparison to uneven or slippery surfaces in real-world conditions. In order to keep the number of strides the same across all participants and all conditions, we could only calculate Lyapunov exponents using 55 strides. Using a greater number of strides, however, could provide estimates of Lyapunov exponents that can better differentiate between conditions [19]. Interpretation of the ICC results could also be somewhat misleading. Because ICC values are calculated as the ratio of between-strides variance to total variance, reliability will increase if between-strides variance increases, even if measurement error is constant [31]. Because ICC values are dependent on the variance of the set of strides being compared, they are most applicable to similar conditions and populations [31]. For example, we found poor reliability for ML MoS from individual conditions (mean ICC 0.31), but moderate reliability (ICC 0.74) when data from all conditions were considered. This result is likely due to greater variance introduced when all data were considered, yielding better ICC coefficients. It is, therefore, important to consider the anticipated variance of the data when deciding whether IMUs will be sufficient to capture trends in walking stability.

Our investigation suggests that IMUs are generally suitable for evaluating gait-stability measures, but may yield poorer performance for ML measures than AP measures compared to optical motion capture. The shortcomings of less accuracy and lower reliability for ML measures could be further exacerbated by certain conditions or between-participant variations, which may have led to less correlation and greater mean differences in our study. Despite these limitations, IMU-derived measures supported the same conclusions as Optical-derived for nearly all tested ML and AP stability measures. Thus, while the measures calculated from IMUs might not be suitable for direct comparisons with optical motion capture, IMUs are capable of detecting changes in gait-stability measures between different conditions and, thus, are promising for evaluating gait stability over the varied terrain of real-world walking conditions.

## Figures and Tables

**Figure 1 sensors-23-01547-f001:**
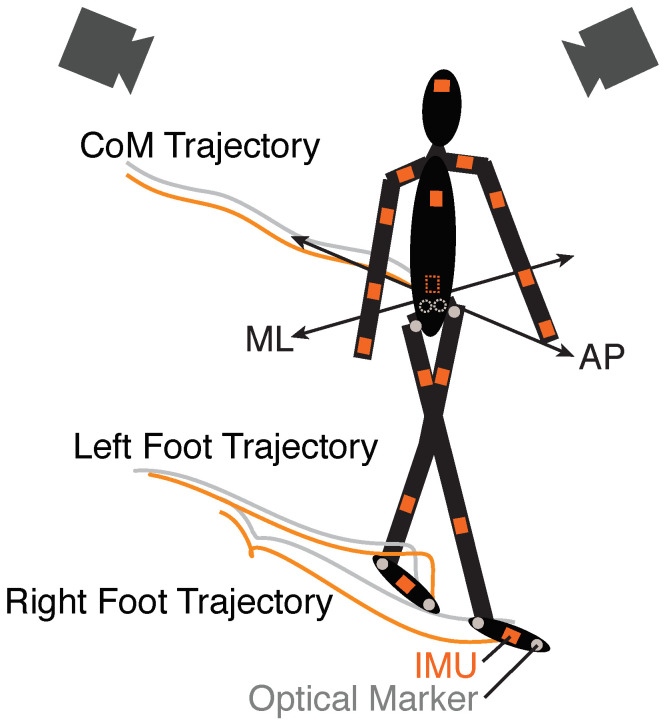
Experimental setup to compare inertial measurement unit (IMU) and Optical measurements. Participants wore an IMU motion-capture suit with seventeen IMUs (orange squares) and eight Optical markers (grey circles) as they walked overground. Exemplar trajectories of the center of mass (CoM) position and each foot are shown for one gait cycle (IMU—orange, Optical—grey). Mediolateral (ML) and anterior-posterior (AP) directions are indicated.

**Figure 2 sensors-23-01547-f002:**
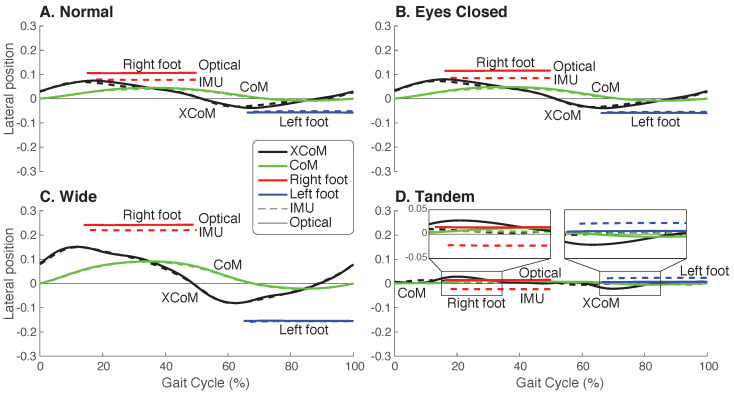
Mediolateral trajectories over a gait cycle for both IMU and Optical. Mean ML XCoM (black), CoM (green), and right (red) and left (blue) BoS for each walking condition of (**A**) Normal, (**B**) Eyes Closed, (**C**) Wide, and (**D**) Tandem, as measured by the IMU (dashed) and Optical (solid) motion-capture systems (N=10). The BoS was determined from lateral foot positions. Positive values indicate the right direction, and negative values indicate the left direction. The gait cycle is shown as a percentage of stride from right heel strike to right heel strike.

**Figure 3 sensors-23-01547-f003:**
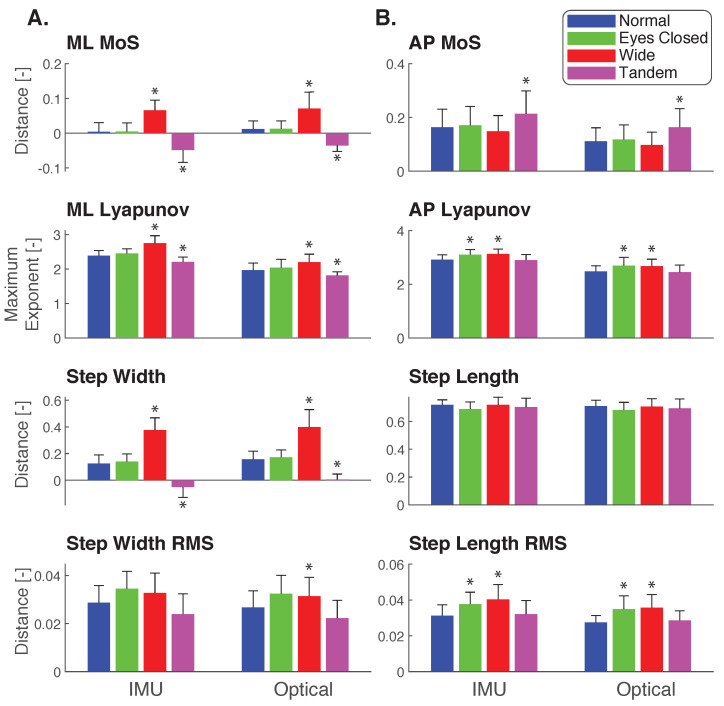
Stability and step placement measures derived from IMU and Optical data. Mean minimum MoS, maximum Lyapunov exponents, step placement, and step placement RMS variability for the (**A**) ML and (**B**) AP directions from IMU (left subcolumn) and Optical (right subcolumn) for Normal, Eyes Closed, Wide, and Tandem. Bars denote averages across all participants (N=10), and error bars denote one s.d. Statistically significant differences from the Normal condition indicated by an asterisk (p<0.05).

**Figure 4 sensors-23-01547-f004:**
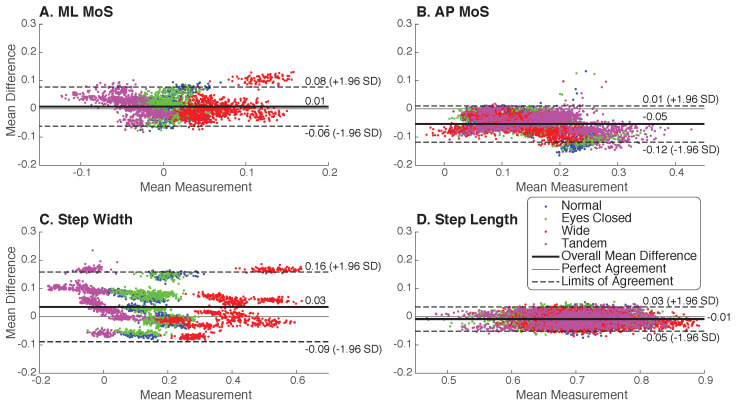
Bland–Altman plots of minimum MoS and step placement measures to compare IMU and Optical results. Mean measurement and mean differences for (**A**) minimum ML MoS, (**B**) minimum AP MoS, (**C**) step width, and (**D**) step length derived from individual strides from all conditions (*N* = 10). Each data point represents an individual gait cycle with mean difference defined as IMU values subtracted from Optical values. Strides from all four conditions were included (Normal—blue, Eyes Closed—green, Wide—red, Tandem—magenta). Overall mean difference (thick solid line) and 95% LoA (dashed line) are shown with the perfect agreement line (thin solid line) for reference.

**Figure 5 sensors-23-01547-f005:**
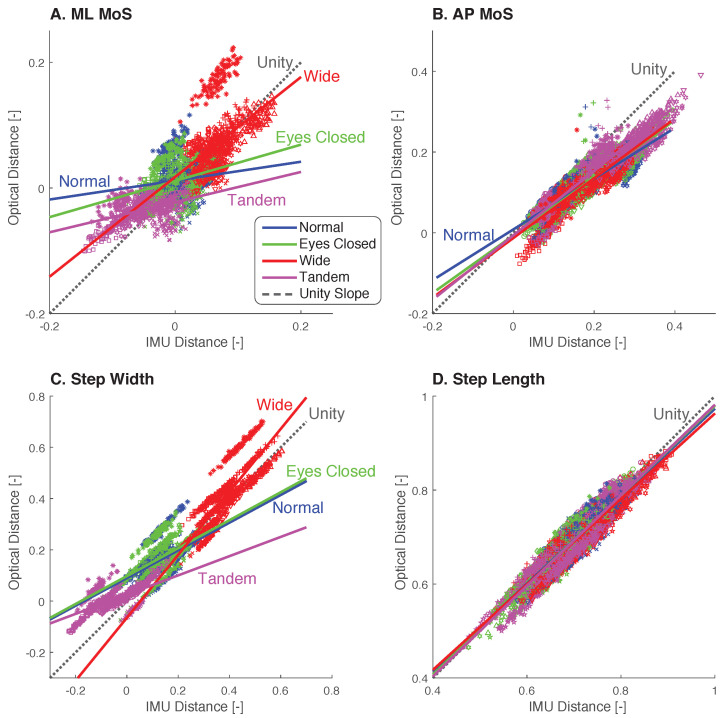
Minimum MoS and step-placement measures with Optical plotted against IMU. Visualization of intraclass correlation (ICC) for (**A**) minimum ML MoS, (**B**) minimum AP MoS, (**C**) step width, and (**D**) step length. Each data point represents an individual gait cycle from each participant (*N* = 10) for all four conditions (Normal—blue, Eyes Closed—green, Wide—red, Tandem—magenta). Linear fits were performed for each condition (solid lines) with perfect agreement indicated by the unity slope (dotted line).

**Table 1 sensors-23-01547-t001:** Summary of quantitative measures and statistical results.

Measure	System	Normal	Eyes Closed	Wide	Tandem	*P*
Speed	IMU	0.357 ± 0.062	0.341 ± 0.070	0.346 ± 0.064	0.316 ± 0.071 ${}^{⋆}$	2.92 × 10${}^{-2}$ *
	Optical	0.354 ± 0.060	0.338 ± 0.069	0.341 ± 0.064	0.313 ± 0.071 ${}^{⋆}$	2.80 × 10${}^{-2}$ *
ML MoS	IMU	0.004 ± 0.026	0.005 ± 0.025	0.066 ± 0.029 ${}^{⋆}$	−0.048 ± 0.036 ${}^{⋆}$	1.49 × 10${}^{-5}$ *
	Optical	0.012 ± 0.023	0.013 ± 0.022	0.071 ± 0.047 ${}^{⋆}$	−0.035 ± 0.017 ${}^{⋆}$	5.93 × 10${}^{-5}$ *
AP MoS	IMU	0.163 ± 0.068	0.170 ± 0.071 ${}^{⋆}$	0.149 ± 0.059 ${}^{⋆}$	0.214 ± 0.085	1.37 × 10${}^{-3}$ *
	Optical	0.111 ± 0.050	0.117 ± 0.055 ${}^{⋆}$	0.097 ± 0.048 ${}^{⋆}$	0.163 ± 0.069	1.57 × 10${}^{-3}$ *
ML Lyapunov	IMU	2.389 ± 0.147	2.455 ± 0.131	2.752 ± 0.216 ${}^{⋆}$	2.209 ± 0.138 ${}^{⋆}$	2.65 × 10${}^{-4}$ *
	Optical	1.969 ± 0.207	2.042 ± 0.242	2.202 ± 0.229 ${}^{⋆}$	1.818 ± 0.099 ${}^{⋆}$	6.60 × 10${}^{-6}$ *
AP Lyapunov	IMU	2.918 ± 0.175	3.096 ± 0.198 ${}^{⋆}$	3.131 ± 0.174 ${}^{⋆}$	2.897 ± 0.207	1.39 × 10${}^{-4}$ *
	Optical	2.474 ± 0.213	2.692 ± 0.309 ${}^{⋆}$	2.674 ± 0.259 ${}^{⋆}$	2.448 ± 0.268	5.30 × 10${}^{-5}$ *
Step Width	IMU	0.029 ± 0.007	0.035 ± 0.007	0.033 ± 0.008	0.024 ± 0.009	3.90 × 10${}^{-2}$ *
RMS	Optical	0.027 ± 0.007	0.032 ± 0.008	0.031 ± 0.008 ${}^{⋆}$	0.022 ± 0.007	3.04 × 10${}^{-2}$ *
Step Length	IMU	0.031 ± 0.006	0.038 ± 0.007 ${}^{⋆}$	0.040 ± 0.008 ${}^{⋆}$	0.032 ± 0.008	6.58 × 10${}^{-3}$ *
RMS	Optical	0.028 ± 0.004	0.035 ± 0.008 ${}^{⋆}$	0.036 ± 0.007 ${}^{⋆}$	0.029 ± 0.005	2.55 × 10${}^{-3}$ *
Step Width	IMU	0.125 ± 0.065	0.140 ± 0.057	0.376 ± 0.092 ${}^{⋆}$	−0.051 ± 0.079 ${}^{⋆}$	5.14 × 10${}^{-6}$ *
	Optical	0.158 ± 0.061	0.172 ± 0.055	0.399 ± 0.132 ${}^{⋆}$	0.004 ± 0.042 ${}^{⋆}$	1.36 × 10${}^{-5}$ *
Step Length	IMU	0.719 ± 0.037	0.688 ± 0.052	0.719 ± 0.055	0.703 ± 0.064	1.13 × 10${}^{-1}$
	Optical	0.711 ± 0.042	0.682 ± 0.056	0.708 ± 0.055	0.694 ± 0.067	1.63 × 10${}^{-1}$

Measures shown as mean ± s.d. across participants (*N* = 10) and reported in dimensionless units. Statistical significance of each measure indicated by an asterisk (*) calculated from repeated measures ANOVA (*P* < 0.05). Statistical significance based on post-hoc t-tests with Holm–Sidak correction with Normal as the control condition indicated by (⋆). MoS, margin of stability; IMU, inertial measurement unit; RMS, root-mean-square.

**Table 2 sensors-23-01547-t002:** ICC values to assess reliability of inertial measurement units (IMU) compared to optical motion capture (Optical). `Overall’ combines data from all conditions.

Measure	Normal	Eyes Closed	Wide	Tandem	Overall
ML MoS	0.16	0.30	0.46	0.34	0.74
AP MoS	0.65	0.72	0.66	0.81	0.74
Step Width	0.51	0.50	0.83	0.42	0.91
Step Length	0.88	0.94	0.93	0.95	0.93

*N* = 10; ICC, intraclass correlation; MoS, margin of stability.

## Data Availability

The data presented in this study are available on request from the corresponding author.

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
