# Peer review of "Validation of Inertial Sensors to Evaluate Gait Stability"

_sensors, 2023, doi:10.3390/s23031547_

Round 1

Reviewer 1 Report

1) 1. The English expression of introduction needs to be optimized. For example, change long sentences into short ones.

2) The interpretation of the superior superior (AP) and medical (ML) needs to be clearer. For example, use a schematic diagram to show.

3) XCoM, BoS, Maximum Lyapunov exponents and other key quantities need to be further explained. It is better to use formulas or schematic diagrams to express them.

4) For ANOVA testing, Bland Altman and ICC,a brief calculation process is required.

Author Response

  1. We have edited the Introduction to shorten long sentences and made other minor grammatical alterations to improve readability. For example, the first Introduction paragraph is now:

Humans encounter a plethora of environmental conditions when walking outside, some of which must be countered to remain upright. Gait analysis has enabled scientific investigations of gait behavior including kinematics and stability measures. However, these studies have predominantly been conducted in controlled laboratory environments with the aid of optical motion capture cameras. While the accuracy of these fixed systems has been well-established [1,2], it is challenging to apply those systems in environments that reflect the diversity of terrain one might experience in everyday life. This disconnect from real-world walking conditions poses a substantial limitation on walking stability and gait adaptation research.

  1. We added ML and AP axes to Figure 1 and included mediolateral (ML) and anterior-posterior (AP) in the figure caption.
  2. The measures we used are well-established for gait stability analysis. We have tried to further expand on them with additional details and references as necessary.
    • The XCoM is defined by equations 1 and 2, and we added to Line 130: The XCoM accounts for both CoM position and velocity… The BoS is defined in Lines 133-138, and we added to Line 130: … created by foot contact with the ground. Both are also shown in Figure 2. Reference 17 provides the foundation of MoS as a stability measure. We added Reference 18, which includes schematics and an additional description of how MoS is applied to human walking research. Reference 18 uses the same terminology as this paper.
    • The use of maximum Lyapunov exponents to assess human walking gait stability is also well-established and thoroughly described in Reference 18. Because the calculation process has been used numerous times in existing literature, we believe it is more practical and relevant to only provide a brief description of how the measure can be interpreted and the specific settings used for our analysis. We also provided citations to a much more thorough description of the algorithm used (Reference 20) and the publicly available code we used (Reference 21). We believe the combination of all these are sufficient to understand and reproduce our analysis.
    • Our other measures of walking speed, step width and variability, and step length and variability are explained in Lines 149 to 155. We added to Line 152-153: Step length was the distance in the AP direction between the two feet at heel strike, and step width was the average ML distance between the two feet during stance.
    • We added Reference 22 for ANOVA/Holm-Sidak correction, a standard statistical method.
    • We added the formula for the Bland-Altman Limits of Agreement (LoA) calculation (Equation 3).
    • Our ICC section mentions the specific type of ICC used and includes references to ICC calculation/interpretation as well as to the publicly available code used. Respectfully, we believe this is sufficient for other researchers to be able to reproduce our results without necessitating the inclusion of standard formulas.

Reviewer 2 Report

I recommend its publication in the journal

Author Response

Thank you for your review.

Reviewer 3 Report

Are all the digits in Table 1 really needed? The table would look much better if the number of digits corresponded to the accuaracy of their evaluation.

The details in Figure 2 are difficult to see. Maybe use different scales for Normal, Wide, and Tandem gaits? The difference in XCoM trajectories of Optical and IMU for all gait types follow the same pattern. Could you comment on this? 

Author Response

  1. We revised Table 1 by reducing significant digits to more accurately reflect accuracy of measurements and maintain consistency to improve table readability.
  2. We added an inset to Tandem subplot on Figure 2 to help show trajectory details. We chose not to alter the y-axis (vertical) scales. A key takeaway for this figure is that ML XCoM and CoM excursions (range) and step width were largest in Wide walking and smallest in Tandem walking and that both IMU and Optical data showed this trend. Observation of the trend requires comparison among the four conditions (subplots) which would be difficult if the scales were different.

We added a comment about the XCoM trajectories to Lines 199 to 201:

The small differences in XCoM trajectories between Optical and IMU were also generally consistent for all conditions but slightly larger for the Tandem condition.